# MocoSFL: enabling cross-client collaborative self-supervised learning

**Jingtao Li**[*]
Arizona State University
Tempe, AZ 85281
jingtao1@asu.edu

**Lingjuan Lyu**
Sony AI
Tokyo, Japan
Lingjuan.Lv@sony.com

**Daisuke Iso**
Sony AI
Tokyo, Japan
Daisuke.Iso@sony.com

**Chaitali Chakrabarti**
Arizona State University
Tempe, AZ 85281
chaitali@asu.edu

**Michael Spranger**
Sony AI
Tokyo, Japan
Michael.Spranger@sony.com

## Abstract

Existing collaborative self-supervised learning (SSL) schemes are not suitable for cross-client applications because of their expensive computation and large local data requirements. To address these issues, we propose MocoSFL, a collaborative SSL framework based on Split Federated Learning (SFL) and Momentum Contrast (MoCo). In MocoSFL, the large backbone model is split into a small client-side model and a large server-side model, and only the small client-side model is processed locally on the client's local devices. MocoSFL is equipped with three components: (i) vector concatenation which enables the use of small batch size and reduces computation and memory requirements by orders of magnitude; (ii) feature sharing that helps achieve high accuracy regardless of the quality and volume of local data; (iii) frequent synchronization that helps achieve better non-IID performance because of smaller local model divergence. For a 1,000-client case with pathological non-IID case (each client only has data from 2 random classes of CIFAR-10), MocoSFL can achieve over 84% accuracy with ResNet-18 model.

## 1 Introduction

Collaborative learning schemes have become increasingly popular, as clients can train their own local models without sharing their private local data. Current collaborative learning applications mostly focus on supervised learning applications where labels are available [Hard et al., 2018, Roth et al., 2020]. However, the fully-labeled assumption may not be practical since labeling can be difficult and requires expertise.

Federated learning (FL) [McMahan et al., 2017] is the most popular collaborative learning framework. One representative algorithm is **"FedAvg"**, where clients send their local copies of the model to the server and the server performs a weighted average operation (weight depends on data proportion) to get a new global model. FL has achieved great success in supervised learning, and has benefited a wide range of applications such as next word prediction McMahan et al. [2017], visual object detection Liu et al. [2020], recommendation Wu et al. [2022a,b], graph-based analysis Chen et al. [2022], Wu et al. [2022c], etc.

---

[*]work done during internship at Sony AI

Workshop on Federated Learning: Recent Advances and New Challenges, in Conjunction with NeurIPS 2022 (FL-NeurIPS'22). This workshop does not have official proceedings and this paper is non-archival.

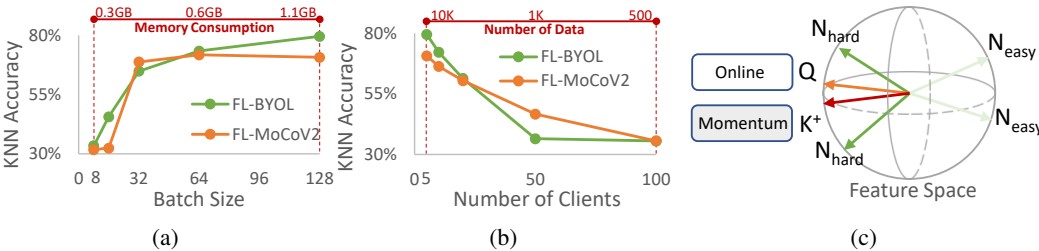

Figure 1: Challenges in FL-SSL schemes. (a) A large batch size is necessary to achieve good performance (KNN validation accuracy [Wu et al., 2018]); (b) Accuracy reduces with an increasing number of clients as a result of insufficient local data; (c) Hard negative keys are essential for the success of contrastive learning.

For more practical collaborative learning on unlabeled data, prior works [Zhang et al., 2020, Zhuang et al., 2021, 2022] combine FL scheme with classic self-supervised learning (SSL) methods such as BYOL [Grill et al., 2020] and Moco [He et al., 2020]. These methods can all achieve good performance when clients' data is Independent and Identically Distributed (IID), however, they suffer from poor performance in non-IID cases. Recently, Zhuang et al. [2022] is able to mitigate non-IID performance drop with divergence-aware aggregation technique and provides the state-of-the-art (SoTA) accuracy performance using a combination of FL and different SSL methods.

However, these SoTA FL-SSL schemes are not practical for cross-client applications, in which number of participants can be thousands even millions, participants' datasets tend to be small, and participants generally possess limited computational power and low technical capabilities [Lyu et al., 2020]. **First**, FL-SSL imposes a significant computation overhead and large memory requirement on clients. This is because SSL requires a large backbone architecture [Chen et al., 2020a] together with a large batch size to ensure good performance. As shown in Fig. 1(a), accuracy drops dramatically when batch size is low for both FL-SSL methods. **Second**, FL-SSL schemes fail to maintain a good accuracy with a large number of clients (cross-client cases), as shown in Fig. 1(b). Given that on a dataset with fixed size, data per client decreases when the number of clients increases, where we notice immediate accuracy degradation. This is mainly because of the failure to meet data requirement in performing contrastive learning.

To solve the above two challenges, we propose MocoSFL, a scheme that adopts Split Federated Learning (SFL) Thapa et al. [2020] as its core and incorporates the feature memory bank and momentum model designs of Moco. We adopt the SFL scheme for three reasons: (i) SFL utilizes a smaller client-side model and so reduces the computation overhead and has lower memory consumption and model parameters; (ii) SFL's latent vector concatenation enables a large equivalent batch size for the centralized server-side model, making the use of micro-batch in training possible for clients, and thus reducing local memory; (iii) SFL's shared server-side model enables effective feature sharing, which removes the requirement of abundant local data and makes the scheme possible for cross-client applications. As a result, our proposed MocoSFL can achieve good accuracy with ultra-low memory requirements and computation overhead and can support a very large number of clients. MocoSFL shows better non-IID performance since local model divergence is smaller.

Our main contributions are:

- We identify two major challenges in FL-SSL schemes for cross-client applications. We find that having a large number of data and a large batch size are two critical factors in achieving a good accuracy performance. However, meeting these requirements is not practical for cross-client applications.

- We propose solutions based on SFL for self-supervised learning. The resulting MocoSFL scheme uses small client-side model, latent vector concatenation, and feature sharing, which effectively solves two challenges, making it amenable for cross-client cases. For cross-silo case, because of less model divergence, the proposed MocoSFL can achieve even better performance than SoTA FL-SSL schemes under non-IID setting.

## 2 Background

### 2.1 Self-Supervised Learning

To learn from unlabeled data, SSL schemes based on contrastive learning such as SimCLR Chen et al. [2020a], BYOL Grill et al. [2020], Simsiam Chen and He [2021] and MoCo He et al. [2020] have achieved great performance on popular benchmarks. We choose MoCo since it uses memory to store negative samples that are absent in other schemes. MoCo relies on InfoNCE loss Oord et al. [2018] as the contrastive mechanism to update its model parameters:

$$\mathcal{L}_{Q,K,N} = -log \frac{exp(Q \cdot K^+/\tau)}{exp(Q \cdot K^+/\tau) + \sum_{N \in M} exp(Q \cdot N/\tau)} \tag{1}$$

where query key $Q$ and positive key $K^+$ are the output vectors of feeding two augmented views of the image to the online model and the momentum model, respectively. $N$ denotes all negative keys in the feature memory $M$. The success of MoCo scheme highly depends on the "hardness" of its negative keys [Kalantidis et al., 2020, Robinson et al., 2020]. The "hardness" for a negative key $N$ in the feature memory bank, can be determined by the similarity (inner-product) between $Q_t$ (at step $t$) and $N$. The smaller the similarity, the better the "hardness" which benefits the optimization process.

MoCo stores current positive keys in a feature memory as negative keys for future iteration, and uses previously stored negative keys to perform contrastive learning. In addition, to maintain the hardness and consistence of negative keys, MoCo adopts a stable momentum model to produce consistent negative keys, that are added to the feature memory at the end of every training step.

### 2.2 Split Federated Learning

SFL Thapa et al. [2020] is a recent collaborative learning scheme that focuses on extreme computation efficiency. It splits the original model architecture into two parts, client-side model copies $C_i$ at clients's local device, and a large server-model $S$ in a cloud server. To complete each training step, clients need to send the latent vectors (the output of client-side model) to the server, and the server processes latent vectors, computes the loss, performs backward propagation and returns the corresponding gradients to clients. Thapa et al. [2020] presents two possible ways for server to process latent vectors sent by clients. In this paper, we use SFL-V1 where the server concatenates all clients' latent vectors and processes them altogether, which makes the batch size equivalently larger at the server and benefits the contrastive learning. In contrast, in SFL-V2, client's latent vectors are processed sequentially in a first-come-first-serve manner that does not bring the "large batch" benefit. We provide detailed process of SFL-V1 in Appendix A.2.

## 3 Motivation

### 3.1 High computing resource requirement

The first challenge is the computing resource requirement of training an SSL model locally. Using a compact backbone model may be accurate enough for supervised learning, but cannot succeed in SSL as it requires a much higher requirement on the model capacity. [Shi et al., 2021, Fang et al., 2021] show that using compact architectures like Mobilenet-V3 [Howard et al., 2019], EfficientNet Tan and Le [2019] suffers from over 10% accuracy degradation compared to a larger ResNet-18 architecture, while an even larger ResNet-50 model is over 15% better over ResNet-18. This means memory requirement for training an SSL model is very high. Using a smaller batch size reduces accuracy dramatically, as shown in Fig. 1(a) and is not an option. A typical FL-SSL scheme costs 590.6 MFLOPs per image and over 1100MB of memory per client for a ResNet-18 with a batch size of 128, which is not practical.

### 3.2 Large data requirement

The other major difficulty for FL-SSL to generalize to the cross-client case is the large data requirement. Especially for cross-client applications, the number of data available to each client can be very limited and biased. For example, a client typically means a hospital in a cross-silo medical

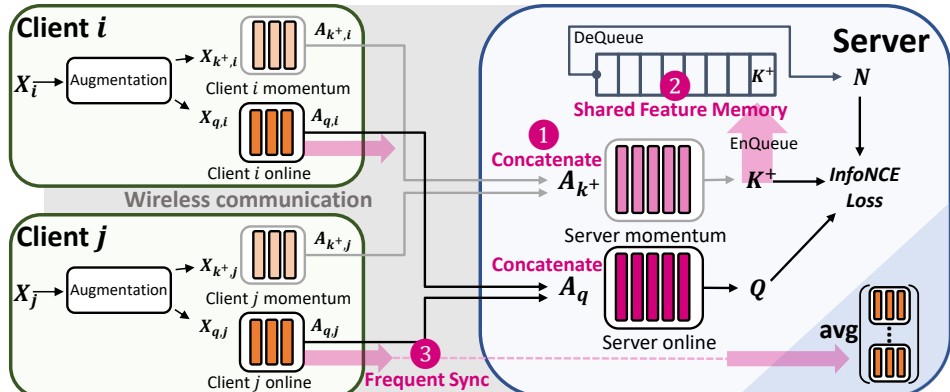

Figure 2: MocoSFL scheme. Three highlighted components are (1) latent vector concatenation, (2) shared feature memory, and (3) frequent synchronization.

application, which has tons of data available. In comparison, in a cross-client application, a client means one patient and thus has limited number of data. The root of the problem lies in the difficulty to find hard negative samples when clients do not have enough local data. And as the number of data is larger, the chance for hard negative samples to present becomes much higher. As a result, existing FL-SSL can only be successful for cross-silo applications where clients can perform effective contrastive learning locally. As demonstrated in Fig. 1(b), we observe high accuracy when clients have 10K samples of data, while the accuracy drops quickly to around 30% for clients having only 500 samples.

## 4 Method

### 4.1 Proposed MocoSFL

Our proposed MocoSFL is an innovative combination of SFL-V1 and MoCo-V2 [Chen et al., 2020b] as shown in Fig. 2. There are three key components to its design. **First**, in each training step, the latent vectors sent by all clients are concatenated before being processed by the server-side model. This helps achieve a large equivalent batch size in order to support mini-batch training. **Second**, we use a shared feature memory which is updated by positive keys contributed by all clients in every training step. **Third**, we use a higher synchronization frequency, which improves the non-IID performance.

### 4.2 Reduce Hardware Resource Requirement

Choice of SFL helps reduce computational overhead and memory consumption because of the much smaller client-side model. For example, on a CIFAR-10 ResNet-18 model with a batch size of 128, a client-side model with 3 layers only costs 13.7% of the FLOPs compared to the entire model, and its memory cost is 227MB, merely one fourth of the entire model. Furthermore, we reduce the batch size to 1 (also known as "micro-batch"), to further reduce the memory consumption. The use of micro-batch in local model training is only possible thanks to the latent vector concatenation mechanism which basically aggregates latent vectors sent by all clients into a big batch before sending it to the server. In addition, in a micro-batch setting, we replace the batch normalization layer by group normalization Wu and He [2018] and weight standardization [Qiao et al., 2019] to gain better accuracy performance. In Fig. 3(b), we compare the computation and memory consumption of the proposed MocoSFL with the FL-SSL scheme. MocoSFL with cut-layer of 3 achieves $\sim 288\times$ reduction in memory consumption than FL-SSL and has 2%-10% higher accuracy. Details of accuracy evaluation are included in Section 5.1.

### 4.3 Mitigate Large Data Requirement

We also find it is necessary to use a large batch size and a large enough feature memory. This finding agrees with the study in Bulat et al. [2021] and also explains the accuracy drop for a small batch

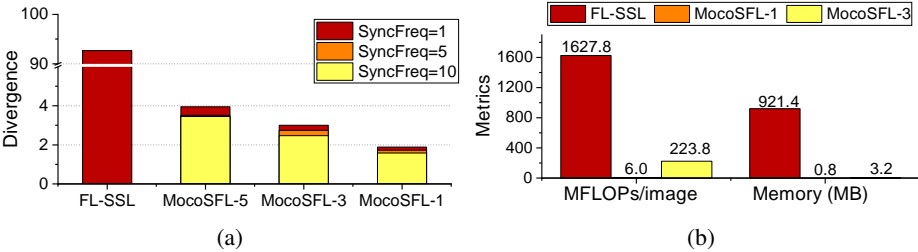

Figure 3: (a) Proposed MocoSFL reduces model divergence. (b) Computation overhead (online model, inference) comparison between FL-SSL scheme and MocoSFL schemes. MocoSFL-$N$: client-side model has $N$ layers.

size in FL-MocoV2 in Fig. 1(a). To illustrate this, consider an ideal case: given that $K_{t-1}$ of size $B$ being the newest negative keys in the feature memory at time $t$, we assume the similarity measure between $K_{t-1}$ and $Q_t$ is a constant $\eta$ for all $t$. We also assume similarity of every negative key batch in feature memory gets reduced by a constant factor $\gamma$ ($\gamma < 1$) after each model update, since model updates degrade similarity. Thus, for a freshly computed query $Q_t$, the total hardness of negative keys in a feature memory can be evaluated as:

$$hardness = B\eta\gamma + B\eta\gamma^2 + ... + B\eta\gamma^{\lfloor M/B \rfloor} \tag{2}$$

$$= B\eta\gamma \times (\frac{1 - \gamma^{\lfloor M/B \rfloor}}{1 - \gamma}) \tag{3}$$

where $B$ is the batch size and $M$ is the feature memory size. We can tell that using a large batch size $B$ is beneficial as it helps maintain better freshness. Also, using a large enough feature memory (increasing $M$) can keep enough negative keys and contribute to a better total hardness. These two requirements are hard to meet in cross-client case because of the memory limit for FL-SSL schemes. Nonetheless, they can be easily fulfilled by MocoSFL because (1) latent vector concatenation enables a large equivalent batch size, and (2) feature memory hosted by the cloud server can be much larger.

### 4.4 Improving Non-IID Performance

We found that use of SFL results in fewer model parameters at the client side and hence smaller model divergence. Furthermore, by introducing frequent synchronization in MocoSFL provides additional reduction in model divergence and greatly improves the non-IID performance. According to Zhuang et al. [2021, 2022], the model divergence can be calculated by:

$$divergence = \frac{1}{EN_C} \sum_{e=1}^{E} \sum_{i=1}^{N_C} \sum_{l=1}^{L} ||W_{e,l}^i - W_{e,l}^*||_2 \tag{4}$$

where $L$ denotes the number of layers for the client-side model, $E$ denotes the total number of synchronizations, $N_C$ denotes the number of clients, and $l, e, i$ are the respective indices for $L, E, N_C$. $W^*$ is the average of all client models $W^i$. MocoSFL reduces the model divergence with two orthogonal mechanisms. The first mechanism is the reduction of client-side model size, which directly results in a lower model divergence. As shown in Fig. 3(a), compared to FL-SSL scheme, MocoSFL has a much lower model divergence when the client-side model has less than 5 layers. The other mechanism is that model synchronization can be done more frequently which helps reduce the model divergence. This is only possible in SFL because the communication overhead is affordable as the client-side model parameter size is much smaller. As shown in the Fig. 3(a), model divergence further reduces as we increase the synchronization frequency.

## 5 Experimental Result

**Experimental Setting.** We simulate MocoSFL clients using different CPU threads and MocoSFL server using a single RTX-3090 GPU. We use ResNet-18 [He et al., 2016] for the majority of the

experiments to better compare with existing SoTA [Zhuang et al., 2022]. We use CIFAR-10 as the main dataset and also present results on CIFAR-100 and ImageNet 12-class subset as in Li et al. [2021]. For the IID case, we assume the entire dataset is divided randomly and equally among all clients. For non-IID experiments, we only consider the pathological (aka. class-wise) non-IID distribution as in McMahan et al. [2017], Zhuang et al. [2022] where we assign only 2 classes of CIFAR-10/ImageNet-12 data or 20 classes of CIFAR-100 data randomly to each client. We perform the MocoSFL training for a total of 200 epochs, using SGD as the optimizer with an initial learning rate of 0.06. For accuracy performance evaluation, we adopt similar linear probe methods as in Grill et al. [2020], Zhuang et al. [2022], where we train a new linear classifier on the representation outputs generated by the MocoSFL backbone model that stays frozen during this process. We leave more details of hyper-parameter choices and evaluations in Appendix A.1.

## 5.1 Accuracy Performance

**Improved non-IID performance.** Fig. 4 shows how using an increased synchronization frequency can significantly improve the non-IID accuracy. We present results for the cut-layer choices of 1 and 3 convolutional layers in the client-side model, represented by "MocoSFL-1" and "MocoSFL-3", respectively. We attribute the improved accuracy to the reduction in model divergence.

**Comparison with FL-SSL.** With synchronization frequency of the MocoSFL scheme set to 10 (per epoch), in cross-silo cases, our proposed MocoSFL achieves significantly better non-IID accuracy performance than Zhuang et al. [2022], despite on the CIFAR-100 dataset when $N_C = 5$, as shown in Table 1.

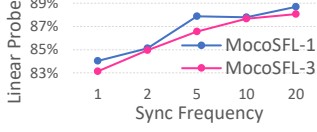

Figure 4: Effect of increasing synchronization

| Method | CIFAR-10 | | CIFAR-100 | |
|---|---|---|---|---|
| | $N_C = 5$ | $N_C = 20$ | $N_C = 5$ | $N_C = 20$ |
| FL-BYOL [Zhuang et al., 2022] | 83.34 | 75.77 | 61.78 | 52.78 |
| MocoSFL-1 (ours) | 87.81 | 85.84 | 58.78 | 57.80 |
| MocoSFL-3 (ours) | 87.29 | 85.32 | 57.70 | 57.52 |

Table 1: Non-IID performance comparison (linear probe accuracy)

**Cross-client Performance.** Our proposed MocoSFL can generalize from a cross-silo application to a cross-client application with 100, 200, and 1000 clients. Note that none of the previous FL-SSL methods can scale to such a large number of clients. Through our experiment, we let each client use a batch size of 1 and an increased synchronization frequency of $f_S = 1000/N_C$, to keep a constant frequency of 50 training steps. We also use a client sampling ratio of $S = 100/N_C$ to keep the same equivalent batch size at the server. The results are shown in Table 2. Note that each client has only 50 data samples in the 1000-client case while the accuracy performance has negligible impact. MocoSFL's accuracy for IID case is almost the same when $N_C$ increases from 100 to 1,000, it has an accuracy drop of only 1% for non-IID case. This small drop is because model divergence scales with number of clients as described in Section 4.4.

Table 2: MocoSFL cross-client accuracy performance (linear probe accuracy) of ResNet-18 model on CIFAR-10, CIFAR-100 and Imagenet-12 datasets with different number of clients $N_C$.

| Method | Dataset | IID | | | non-IID | | |
|---|---|---|---|---|---|---|---|
| | | $N_C = 100$ | $N_C = 200$ | $N_C = 1000$ | $N_C = 100$ | $N_C = 200$ | $N_C = 1000$ |
| MocoSFL-1 | CIFAR-10 | 87.29 | 87.38 | 87.51 | 87.71 | 87.39 | 86.46 |
| | CIFAR-100 | 58.91 | 59.15 | 58.85 | 59.22 | 58.90 | 56.75 |
| | ImageNet-12 | 92.02 | 91.73 | 91.76 | 92.24 | 91.44 | 91.28 |
| MocoSFL-3 | CIFAR-10 | 87.29 | 87.15 | 87.25 | 87.10 | 85.22 | 84.09 |
| | CIFAR-100 | 58.41 | 58.30 | 58.80 | 58.69 | 58.59 | 56.88 |
| | ImageNet-12 | 92.08 | 92.24 | 92.02 | 92.60 | 91.83 | 91.28 |

# 6 Conclusion

We propose MocoSFL, a collaborative SSL framework based on SFL. The proposed framework addresses hardware resource requirement at client-side by enabling small batch size training and computation offloading. It also relieves the large data requirement of local contrastive learning by enabling effective feature sharing. The proposed scheme can support massive number of clients. More importantly, we achieve even better IID/non-IID performance with much lower hardware requirement than the SoTA FL-based SSL methods.

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

# A Technical Details

## A.1 Details of Hyper-parameter Choices

**Training Hyper-parameters.** For the data preprocessing, we split each dataset into 50K training and 10K validation data. And we use the training data directly as query data $X_q$, and its augmented version as positive data $X_{k+}$. We use the same augmentation pipeline as in Chen et al. [2020b]. For the backbone model, we remove the last fully-connect layer and replace it with two larger fully connected layers of 1024 neurons as the MLP head, to fit well with small batch size training, we replace the batch normalization layers by *group normalization* layers and set the number of channel per group to 4 (while keeping the minimum number of group to 1, if input channels is less than 4). We also use the weight standardization technique Qiao et al. [2019] in combination with *group normalization*, which is reported to perform better. At the server-side, we use a feature memory of 6,000 negative keys that is implemented as a First-In-First-Out (FIFO) queue, and we update it by pushing newly generated positive keys at every training step. For contrastive loss calculation, we use the symmetric contrastive loss calculation introduced in Moco-V3 [Chen et al., 2021], where the position of query and positive keys are swapped in Eq. (1), and the new loss takes the form of $(\mathcal{L}_Q, K, N + \mathcal{L}_K, Q, N)/2$. For the optimization process, we update the online model of both client-side and server-side model with the SGD optimizer with 0.06 initial learning rate, 0.9 momentum and 0.0005 weight decay. We schedule the learning rate using the *cosine annealing* trick for both client-side and server-side models. The momentum model update is done through the moving average mechanism in He et al. [2020]. In the macro level, we set the number of epochs to a fixed of 200 epochs, where we observe the performance almost converges. To be noticed, according to previous study Chen et al. [2020b], the training accuracy can increase with extensive long training (i.e. >800 epochs). Instead of using a long training session, we set it to 200 epochs to maximize time & performance tradeoff. At the end of each epoch, we perform a k-nearest-neighbor (knn) validation using the 20% validation dataset, and finally use the best of which to perform the final linear probe evaluation.

**Evaluation Hyper-parameters.** For the knn validation, we feed the validation dataset and use the normalized feature output of the backbone (together with the MLP head) and follows the implementation in Wu et al. [2018] to get the accuracy. For the linear probe evaluation, we follow exactly the same procedure as in Zhuang et al. [2022], where we replace the MLP head with a randomly initialized fully-connected layer, fine-tune the new model on the labeled dataset while keeping the backbone parameters frozen. We train the single fully-connected layer for a total of 100 epochs on the labeled training dataset and use the accuracy on the labeled validation dataset as final accuracy. To optimize the single FC layer, we use a batch size of 128 and Adam optimizer with default learning rate 0.001 and use cosine annealing to schedule its learning rate.

**FL Hyper-parameters: Synchronization Frequency and Client Sampling Ratio.** In our proposed MocoSFL scheme, we present an auto-adjust mechanism on adjusting the synchronization frequency, the client-sampling ratio, and the local batch size according to number of clients, which is supplied by the scheme owner. We use it for all of our results for MocoSFL. The design of the auto-adjust is to strike a balance between model divergence and sample hardness and is based on two principles. **The first principle** is to always keep the equivalent batch size greater than 100, so that a good hardness can be kept (indicated by Eq. (3)). For example, if only 5 clients join the MocoSFL, we force each client to have a batch size of 20 to keep the equivalent batch size to 100. If 1,000 clients join the MocoSFL, we use a client sampling ratio of 0.1 to let 100 clients join in each round. **The second principle** is the synchronization frequency must be adjusted correspondingly to the number of local training steps. The reason is that more local updates leads to higher model divergence, which needs to be mitigated using more frequent synchronization. = For example, for a 1,000-client MocoSFL where each client has 50 data and perform 50 local training steps each epoch (with batch size of 1), the default synchronization frequency of "1/epoch" can suffice and we do not need to increase it. However, for a 100-client MocoSFL where each client performs 500 local training steps, accuracy drops for a frequency of "1/epoch", so we increase the synchronization frequency to "10/epoch".

## A.2 Detailed SFL-V1 Process

The training process of SFL-V1 in supervised learning is shown in Algorithm 1. Since we find the large equivalent batch size essential for the success of contrastive learning, we adopt SFL-V1 since a large equivalent batch size (equal to the number of clients times clients' batch size) can be achieved

**Algorithm 1** Split Federated Learning V1 [Thapa et al., 2020]

**Require:** For $N$ clients, instantiate private training data $(\mathbf{X}_i, \mathbf{Y}_i)$ for $1, 2, ..., N$. Each client-side model $C_i$ has $M$ layers and server-side model $S$ has $L - M$ layers.

1: **for** epoch $t \leftarrow 1$ to num_epochs **do**
2:    $C^* = \frac{1}{N}\sum_{i=1}^{N} C_i$;  $C_i \leftarrow C^*$ for all $i$               {Model Synchronization}
3:    **for** step $s \leftarrow 1$ to num_batches **do**
4:       **for** client $i \leftarrow 1$ to $N$ **in Parallel do**
5:          data batch $(\boldsymbol{x}_i, \boldsymbol{y}_i) \leftarrow (\mathbf{X}_i, \mathbf{Y}_i)$
6:          $\boldsymbol{A}_i = C_i(\boldsymbol{W}_{C_i}; \boldsymbol{x}_i)$                 {Send $\boldsymbol{A}_i$ to Server}
7:       **end for**

8:       $\boldsymbol{A} = cat(\boldsymbol{A}_1, \boldsymbol{A}_2, ..., \boldsymbol{A}_N)$;  $\boldsymbol{y} = cat(\boldsymbol{y}_1, \boldsymbol{y}_2, ..., \boldsymbol{y}_N)$
9:       $\mathcal{L} = \mathcal{L}_{CE}(S(\boldsymbol{W}_S; \boldsymbol{A}), \boldsymbol{y})$
10:      $\nabla_{\boldsymbol{A}}\mathcal{L} \leftarrow$ back-propagation     {Partition $\nabla_{\boldsymbol{A}}\mathcal{L}$, Send $\nabla_{\boldsymbol{A}_i}\mathcal{L}$ Correspondingly to Client $i$}
11:      Update $\boldsymbol{W}_S$;

12:      **for** client $i \leftarrow 1$ to $N$ **in Parallel do**
13:         $\nabla_{\boldsymbol{x}_i}\mathcal{L} \leftarrow$ back-propagation
14:         Update $\boldsymbol{W}_{C_i}$;
15:       **end for**
16:    **end for**
17: **end for**

thanks to the latent vector concatenation in line 8. Difference between SFL-V1 and V2: V1 performs concatenation on clients' latent vectors and feeds it to S and update the model for only one time (call "optimizer.step() for one time" in pytorch), while V2 sequentially feeds them and updates S for a total of $N$ ($N$ : number of clients) times (call "optimizer.step() for $N$ time").

## B   Extensive Empirical Results

In this section, we perform extensive empirical evidence of the proposed MocoSFL scheme on different architecture (Appendix B.1) and performance of MocoSFL in a semi-supervised learning application (Appendix B.2).

### B.1   ResNet-50 results

We follow the same setup in Table Table 1 and Table 3 on ResNet-50 architecture for cross-silo (5 and 20 clients) and cross-client setting (100 clients). As shown in Table 3, compared with ResNet-18 model, using a larger model brings $\sim$3% better accuracy performance.

| Method | $N_C = 5$ | | $N_C = 20$ | | $N_C = 100$ | |
|---|---|---|---|---|---|---|
| | IID | Non-IID | IID | Non-IID | IID | Non-IID |
| MocoSFL-1 (ResNet-18) | 87.38 | 87.81 | 87.21 | 85.84 | 87.29 | 87.71 |
| MocoSFL-3 (ResNet-18) | 87.09 | 87.29 | 87.09 | 85.32 | 87.29 | 87.10 |
| MocoSFL-1 (ResNet-50) | 90.54 | 90.29 | 90.27 | 88.78 | 90.17 | 90.37 |
| MocoSFL-3 (ResNet-50) | 90.86 | 90.94 | 90.58 | 89.83 | 90.67 | 90.66 |

Table 3: MocoSFL Performance (linear probe accuracy) on a ResNet-50 model

### B.2   Semi-supervised Learning Evaluation

We report the performance of using the proposed MocoSFL for semi-supervised learning application for both cross-silo (5 and 20 clients) and cross-client setting (100 clients). As shown in Table 4, the proposed MocoSFL achieves competitive accuracy performance. For ResNet-18 model, we achieve

10% higher accuracy than FL-BYOL [Zhuang et al., 2022] if only 1% label is given on a CIFAR-10 dataset.

| Arch | Method | $N_C = 5$ | | $N_C = 20$ | | $N_C = 100$ | |
|---|---|---|---|---|---|---|---|
| | | 1% | 10% | 1% | 10% | 1% | 10% |
| **ResNet-18** | FL-BYOL [Zhuang et al., 2022] | 73.44 | 79.49 | N/A | N/A | N/A | N/A |
| | MocoSFL-1 | 84.69 | 87.05 | 76.80 | 84.46 | 82.27 | 86.84 |
| | MocoSFL-3 | 84.46 | 86.50 | 76.70 | 84.20 | 81.72 | 85.91 |
| **ResNet-50** | FL-BYOL [Zhuang et al., 2022] | 72.52 | 80.68 | N/A | N/A | N/A | N/A |
| | MocoSFL-1 | 86.28 | 89.55 | 80.34 | 86.09 | 84.83 | 88.78 |
| | MocoSFL-3 | 87.09 | 90.33 | 81.59 | 87.21 | 84.65 | 88.81 |

Table 4: MocoSFL Semi-supervised learning Performance (Non-IID).

