# OpenReview forum: "MocoSFL: enabling cross-client collaborative self-supervised learning"
_NeurIPS.cc/2022/Workshop/Federated_Learning — FL-NeurIPS 2022 Poster_

### Official Review · Reviewer_VJob · 2022-10-17
**Draft with good quality, but slightly lack of novelty**

This paper proposes MocoSFL, a method that seeks to solve self-supervised federated learning. The authors identified two major issues of federated self-supervised learning, i.e., high computing and resource requirements and large data requirements. To tackle such issues, MocoSFL is designed leveraging latent embedding concatenation, feature sharing, and frequent synchronization. Experimental results are given to demonstrate the performance of MocoSFL.

Pros:
- The paper is well written.
- Solving federated self-supervised learning is a promising research direction to pursue.

Cons:
- The proposed method is very similar to [1]. Though [1] focuses mainly on supervised learning tasks, but its technical cores are very similar to MocoSFL.
- I'm not fully convinced that the large data requirements are necessary in practice. Overall, I believe in the real world there is no concept "dataset", and each client's dataset is only determined by how many data points each user owns. And thus a larger number of clients does not necessarily mean a smaller per-client number of data points.
- Usually, communication is more expensive than computation in the federated learning scenario (e.g., wireless communication usually enjoys much lower bandwidth etc). Thus does the frequent communication method in MocoSFL lead to a significant slow down in the federated learning process?

[1] https://proceedings.neurips.cc/paper/2020/hash/a1d4c20b182ad7137ab3606f0e3fc8a4-Abstract.html

---

### Official Review · Reviewer_uCNs · 2022-10-18
**Interesting approach to enabling self-supervised learning on constrained clients**

The authors propose a split learning approach for enabling self-supervised learning (SSL). Of particular concern is the size of model and number of samples required to successfully execute SSL. The authors propose a reduction of model size by moving to a split paradigm, whereby only the first N layers are trained by a client (N depending on memory capacity) while the rest of training is completed by the server. Concatenation of several latent outputs over a batch of inputs simulates microbatching alleviating computational and memory overhead.

Pros:
-Creative use of split learning. Federated SSL is still an emerging subfield and this is, to the best of my knowledge, a unique approach.
-Convincing empirical results.
-Flexible to client memory constraints. The cut layer can be dynamically determined based on client capacity.

Cons:
-The shared memory mechanism for holding positive and negative keys is confusing and needs better exposition. My primary concern is that this shared memory may somehow harm privacy, especially if a global dataset is known. Please address why this shared memory is not an issue from a privacy perspective. This is the primary reason for not assigning a higher score.
-Seems odd that placement of the cut layer at 1 performs better than the cut at 3, i.e., more embedding layers. Please discuss how placement of the cut affects performance.

---

### Decision · Program_Chairs · 2022-10-20

Accept (Poster)